# IT-NAS: Integrating Lite-Transformer into NAS for Architecture Seletion

## Abstract

Neural Architecture Search (NAS) aims to search for the best network in the pre-defined search space. However, much work focuses on the search strategy but little on the architecture selection process. Despite the fact that the weight-sharing based NAS has promoted the search efficiency, we notice that the architecture selection is quite unstable or circuitous. For instance, the differentiable NAS may derive the suboptimal architecture due to the performance collapse caused by bi-level optimization, or the One-shot NAS requires sampling and evaluating a large number of candidate structures. Recently, the self-attention mechanism achieves better performance in terms of the long-range modeling capabilities. Considering that different operations are widely distributed in the search space, we suggest leveraging the self-attention mechanism to extract the relationship among them and to determine which operation is superior to others. Therefore, we integrate Lite-Transformer into NAS for architecture selection. Specifically, we regard the feature map of each candidate operation as distinct patches and feed them into the Lite-Transformer module along with an additional Indicator Token (called IT). The cross attention among various operations can be extracted by the self-attention mechanism, and the importance of each candidate operation is then shown by the softmax result between the query of indicator token (IT) and other values of operational tokens. We experimentally demonstrate that our framework can select the truly representative architecture in different search spaces and achieves 2.39% test error on CIFAR-10 in DARTS search space, and 24.1% test error on ImageNet in the ProxylessNAS (w/o SE module) search space, as well as the stable and comparable performance in NAS-Bench-201 search space, S1-S4 search spaces and NAS-Bench-1Shot1 search space.

## 1 Introduction

Neural Architecture Search (NAS) is emerging as a new paradigm for designing network structures. It has been demonstrated to outperform manually designed networks in many tasks, including image classification (Zoph & Le, 2016; Zoph et al., 2018; Guo et al., 2020), object detection (Chen et al., 2019; Ghiasi et al., 2019), semantic segmentation (Chen et al., 2018; Liu et al., 2019) and so on. The fundamental disadvantage of earlier NAS methods, which primarily relied on heuristic algorithms like reinforcement learning (Baker et al., 2016; Bello et al., 2017; Zoph et al., 2018) or evolutionary algorithms (Real et al., 2017; Liu et al., 2018a; Real et al., 2019), is the necessity to train each architecture from scratch for validation, thereby impeding further advancement of NAS.

Fortunately, ENAS (Pham et al., 2018) proposes a weight-sharing mechanism, which greatly improves search efficiency. More recently, the differentiable NAS methods (Liu et al., 2018b; Xie et al., 2018) and the One-shot NAS methods (Guo et al., 2020; Chu et al., 2021b; You et al., 2020) have been more popular. They both firstly train a super-network, and then derive the final architecture based on different strategies. However, the differentiable approach selects the target network with the largest architecture parameters, which cannot fully reflect the true operation strength (Wang et al., 2021a; Xie et al., 2021b). Nevertheless, DARTS-PT requires fine-tuning after discretizing each edge, resulting in additional selection time. The One-shot method selects the optimal one by sampling and evaluating a large number of candidate structures, yielding the time-consuming issues (Chen et al., 2021a). Whereas, BN-NAS is not suitable for search space without batch normaliza-

tion layer. We are endeavoring to explore whether there is a more appropriate way for both popular search spaces to robustly and quickly select the architecture, as has rarely been studied before.

The attention mechanism can be used to emphasize the important components of the input while ignoring other trivial ones. The Transformer architecture (Vaswani et al., 2017) has reignited a boom in the field of Computer Vision (CV) (Touvron et al., 2021; Liu et al., 2021) since ViT (Dosovitskiy et al., 2021) achieves competitive performance compared to Convolutional Neural Networks (CNNs). It simply slices the image into small patches and then model the cross-attention between long sequences to locate key information, resulting in better performance. Meanwhile, Batchformer (Hou et al., 2022) introduces a batch transformer module that is applied to the batch dimension of each mini-batch of data to implicitly explore the sample relationships. Inspired by this, we can intuitively analogize the candidate operations in the NAS search space to patches, and then leverage the self-attention mechanism to describe the interactions among different operations. As a result, the optimal architecture can be selected according to the self-attention weights.

In summary, we suggest integrating Lite-Transformer into NAS for architecture selection. Specifically, we insert the Lite-Transformer module on each edge and determine the optimal operation associated with that edge in the cell-based search space. While in the chain-style search space, where the network is defined by a sequence of layers containing various choice blocks, the Lite-Transformer module is inserted on each layer to select the appropriate block. We adopt a more broad perspective that treats candidate operations as patches. The patches are then linearly mapped and further packed into three matrices, namely Q, K, and V. The softmax result between Q and K is a square matrix termed the attention map, which can be regarded as the attention weight between different candidate operations.

Furthermore, to address the issue of asymmetric attention matrix, i.e., the mutual attention value between any two operations cannot determine which is more favorable. We introduce an additional indicator token (called IT) to calculate the cross-attention between IT and the other operational tokens, which is inspired by the truth claimed by EViT (Liang et al., 2022) that the class token can be used to determine the importance of other tokens. In this way, the importance corresponding to each operation can be represented by the row of the indicator token (IT) in the attention matrix. After the super-network along with the Lite-Transformer module is trained to convergence, we simply need to forward propagate once on the validation dataset in order to determine the optimal architecture by computing the self-attention weights based on the indicator token (IT).

In general, our main contributions can be summarized as follows:

- We revisit the architecture selection process of neural architecture search (NAS) in a fresh perspective and, to our knowledge, are the first to integrate Lite-Transformer into NAS for architecture selection, utilizing the self-attention mechanism to explore the interaction among different candidate operations by regarding each one as operational token.

- We introduce an additional indicator token (called IT) to compute the cross-attention between IT and the other operational tokens. In this case, the row of IT in self-attention weights matrix can be used to establish the priority of each candidate operation.

- Experimental results show that IT-NAS achieves better performance in DARTS and ProxylessNAS search space, as well as stable and comparable performance in NAS-Benches, including S1-S4, NAS-Bench-201, and NAS-Bench-1Shot1.

- More comprehensive experiments demonstrate the robustness and effectiveness of IT-NAS in selecting architectures. We also theoretically and empirically analyze why the self-attention mechanism can effectively select optimal architectures, proving the priority of our proposed method.

## 2 RELATED WORKS

**Neural Architecture Search.** Neural Architecture Search (NAS) aims to select the optimal architecture in the pre-defined search space. Earlier NAS approaches (Baker et al., 2016; Zoph et al., 2018; Real et al., 2017; 2019) incurred substantial search overhead due to the requirement to train each candidate architecture from scratch. ENAS (Pham et al., 2018) firstly proposed the weight-sharing mechanism such that weights can be shared among different sub-structures in the super-network,

greatly reducing the validation time. Based on the mechanism, differentiable NAS and One-shot NAS became the two most mainstream paradigms. Thereafter, a series of subsequent works focused on the design of search strategies and the improvement of super-network training.

The differentiable NAS (Liu et al., 2018b) introduced architecture parameters which are mainly trained alternately with the network weights. However, the search process is very unstable and suffers from the performance collapse issue. To improve the robustness of the search process, RobustDARTS (Zela et al., 2020) and SDARTS (Chen & Hsieh, 2020) analyzed and fixed optimization errors from a mathematical point of view. To reduce the discretization gap of architecture selection, SNAS (Xie et al., 2018), GDAS (Dong & Yang, 2019), DATA (Chang et al., 2019) leveraged reparameterization techniques, and FairDARTS (Chu et al., 2020) introduced auxiliary loss, all with the aim of optimizing architecture parameters to approximate discrete forms. These prior methods select target network based on architecture parameters, but recent work Wang et al. (2021a) pointed out that the operation associated with the largest magnitude of architecture parameters does not necessarily result in the highest validation accuracy after discretization. It proposed a perturbation-based architecture selection strategy, which, however, requires fine-tuning after discretizing each edge, leading to substantial computation costs.

The One-shot NAS methods (Guo et al., 2020) focused on the super-network training. SPOS (Guo et al., 2020) proposed to sample and train single-path sub-network uniformly; FairNAS (Chu et al., 2021b) guaranteed all paths are sampled each time for fairness; RLNAS (Zhang et al., 2021) focused on training with random labels. These methods then select the optimal substructure by sampling many sub-networks based on the evolutionary algorithm to evaluate their performance, which is very time-consuming. BN-NAS (Chen et al., 2021a) proposed to train only BN layers and then select the architecture according to the parameters of BN layer. But it is obvious that this method may not be effective for some networks lacking BN layer in the search space, which limits its application.

**Vision Transformer.** Transformer (Vaswani et al., 2017) has drawn much attention to computer vision recently due to its strong capability of modeling long-range relations. ViT (Dosovitskiy et al., 2021) firstly introduced the pure Transformer Encoder without any convolutional layers and achieved comparable performance to CNNs. From then on, many works attempted to modify the ViT architecture to image classification (Yuan et al., 2021; Zhou et al., 2021), object detection (Carion et al., 2020; Zhu et al., 2020; Liu et al., 2021), and semantic segmentation (Wang et al., 2021b; Xie et al., 2021a; Chu et al., 2021a).

To further improve the performance of Transformer-like network, some works combine with NAS to automatically discover better architecture. For example, AutoFormer (Chen et al., 2021c) firstly aimed to search ViT. GLiT (Chen et al., 2021b) introduced locality modules into the search space and searched ViT from both global and local levels. ViT-ResNAS (Liao et al., 2021) proposed to search for multi-stage ViT architecture. Besides, other approaches make ViT more efficient by reducing the number of tokens. DynamicViT (Rao et al., 2021) introduced an additional learnable neural network to reduce the tokens of a fully trained ViT, TokenLearner (Ryoo et al., 2021) aggregated the entire feature map weighted by a dynamic attention map, and EViT (Liang et al., 2022) focused on the progressive selection of informative tokens during training.

Unlike searching the ViT network, we leverage Transformer to boost NAS. Inspired by the fact that the number of tokens can be reduced with little impact on performance, we analogize the candidate operations to tokens, and then use self-attention mechanism to select the truly important operations.

## 3 METHODOLOGY

### 3.1 NAS OVERVIEW

**Differentiable NAS.** In the traditional cell-based search space, each cell is defined as a directed acyclic graph (DAG) with $N$ nodes, and each edge $(i, j)$ between every node is associated with mixed operation $\bar{o}^{(i,j)}$ that is parameterized as architecture parameters $\alpha^{(i,j)}$ by using softmax relaxation. The differentiable architecture search can be formulated as a bi-level optimization problem:

$$
\begin{aligned}
\min_{\alpha} \quad & \mathcal{L}_{val}(w^*(\alpha), \alpha) \\
s.t. \quad & w^*(\alpha) = \arg\min_w \mathcal{L}_{train}(w, \alpha)
\end{aligned}
\tag{1}
$$

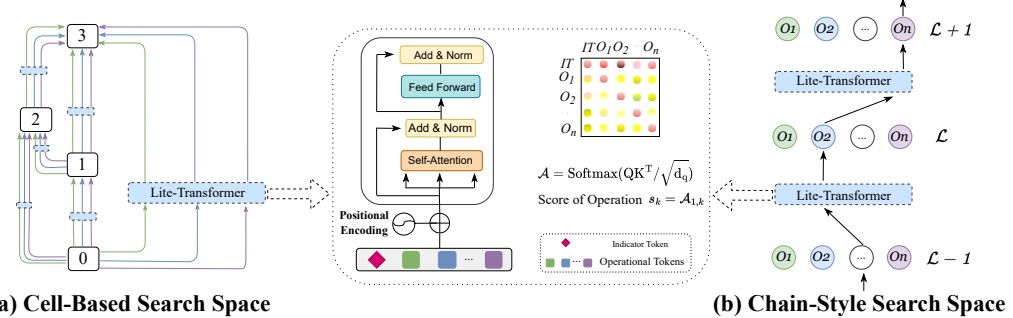

**(a) Cell-Based Search Space**                **(b) Chain-Style Search Space**

Figure 1: The main framework of IT-NAS. Specifically, the candidate operations on each searchable layer are regarded as operational tokens, added position encoding, and fed into the Lite-Transformer module. The Indicator Token (IT) is then introduced to extract the relationship between different operations and assign attention weights to them for selecting the optimal architecture.

The optimal cell architecture is selected according to the architecture parameters $\alpha$ by discretizing each edge. In this way, the differentiable NAS methods argue that the optimized architecture parameters can represent the true importance of the candidate operation, which is unfortunately not the case (Wang et al., 2021a; Xie et al., 2021b).

**One-shot NAS.** In the chain-style search space, One-shot NAS methods mainly focus on super-network training, which is expressed as:

$$w^*(a) = \arg\min_w \mathbb{E}_{a \sim \mathcal{A}} \mathcal{L}_{train}(w, a) \tag{2}$$

Compared to Eq.(1), the continuous architecture parameters are discarded, and only network weights are optimized. So the architecture selection process needs to sample a large number of sub-networks by evolution algorithm and evaluate each of them by sharing the super-network weights, resulting in the time-consuming issues (Guo et al., 2020; Chen et al., 2021a).

No matter in the cell-based search space or the chain-style search space, any two network layers are composed of all candidate operations, with the goal of selecting out the optimal one from them. We take a new perspective for architecture selection, that is, neither using architecture parameters nor sampling sub-networks, but leveraging the self-attention mechanism to evaluate which operation is superior, and this has never been studied in NAS before.

## 3.2 SELF-ATTENTION MECHANISM

To model the relationship between individual different candidate operations, we first regard them as patches and then package all candidate operations of the layer into a sequence and feed them to the Lite-Transformer Encoder.

Let's denote the feature maps sequence of $n$ candidate operations on the searchable layer $\mathcal{L}$ as $\mathcal{O}_{\mathcal{L}}^i \in \mathbb{R}^{b \times c \times h \times w}(i = 1, 2, ..., n)$. We first flatten each feature to $\mathcal{O}_{\mathcal{L}}^i \in \mathbb{R}^{b \times 1 \times (c*h*w)}$ so that each one can be analogized as a patch similar to ViT (Dosovitskiy et al., 2021). And then we concatenate $n$ feature maps in the second dimension as $\mathcal{O}_{\mathcal{L}} \in \mathbb{R}^{b \times n \times (c*h*w)}$. For shorthand, let $d = c * h * w$ denotes the embedding dimension.

**Self-Attention.** A transformation layer projects each operational sequence $X \in \mathbb{R}^{n \times d}$ to three different sequential vectors (namely, query $Q$, key $K$, and value $V$), where $n$ and $d$ are the length and dimension of the input sequences, respectively. The two sequences of query and key input scheme are referred to as cross-attention mechanism. Specifically, it explicitly aggregates the query with the corresponding key, assigns them to the value, and updates the output vector:

$$\text{SelfAttention}(Q, K, V) = \text{Softmax}(\frac{QK^T}{\sqrt{d_q}})V \tag{3}$$

where $d_q$ is the length of query vector. The result of $\text{Softmax}(\cdot)$ is the self-attention weight square matrix, where each entry represents the cross-attention between any two query and key. However,

this matrix is not symmetric as shown in Figure 1. That is, the attention weight between query $O_i$ and key $O_j$ is not equal to the attention weight between query $O_j$ and key $O_i$, so it is not possible to directly measure whether $O_i$ or $O_j$ is more important.

**Indicator Token.** We introduce an additional indicator token (called IT), which aims to explicitly indicate the importance of each candidate operation. The IT is concatenated in series in front of the operational tokens before feeding them into the Lite-Transformer Encoder. The interaction between IT and other operational tokens is computed by the self-attention mentioned before:

$$Z_{it} = \text{Softmax}(\frac{q_{it}K^T}{\sqrt{d_q}})V = a \cdot V \tag{4}$$

where $q_{it}$ is the query of indicator token, $K$ and $V$ are operational key matrix and value matrix. So, $Z_{it}$ denotes the linear combination of the value vectors $V = [v_1, v_2, ..., v_n]$ with the attention weights vector $a = [a_1, a_2, ..., a_n]$. Because $v_i$ is the linear transformation from $i$-th operation, the attention weight $a_i$ therefore can be seen how importance of this operation over other operations.

### 3.3 ARCHITECTURE SELECTION

**Supernet Training.** The Lite-Transformer Encoder is simplified to repeat only once and consists of a single-head Self-Attention (SA) and a Feed-Forward-Network (FFN). Residual connections alongside layer normalization (LN) (Ba et al., 2016) are employed after each layer. We integrate Lite-Transformer $\mathcal{T}$ into super-network $\mathcal{N}$, and then train them following the settings same as DARTS (Liu et al., 2018b) for differentiable NAS or SPOS (Guo et al., 2020) for One-shot NAS. Therefore, Eq.(1) and Eq.(2) can be unified as follows:

$$\mathcal{W}_N^* = \arg\min_{\mathcal{W}_\mathcal{N}} \mathcal{L}_{train}(\mathcal{N}(\mathcal{W}_N), \mathcal{T}(\mathcal{W}_T; \mathcal{W}_N)) \tag{5}$$

where $\mathcal{W}_\mathcal{N}$ denotes the super-network weights and $\mathcal{W}_T$ is the Transformer wights which is optimized together with the super-network by minimizing losses on the training dataset. As a result, we unify the differentiable NAS and One-shot NAS into a mono-level optimization. The indicator token (IT) can access the knowledge about the effect of different operations to the network during this period by exploring the interaction among candidate operations.

**Architecture Selection.** The architecture selection process is performed after the training is completed. We propagate all images of the validation datasets to obtain the self-attention weights. Specifically, the first row of the result of the $\text{Softmax}(\cdot)$ matrix, i.e., the cross-attention between the query of the indicator token (IT) and the keys of other operational tokens, denotes the importance of each candidate operation. In this case, we can select the optimal operation on each searchable layer according to this importance indicator:

$$\begin{aligned} A^* &= argmax(a[1:]) \\ &= argmax([a_1, a_2, ..., a_n]) \end{aligned} \tag{6}$$

Because $a_0$ indicates the self-attention of the query of indicator token (IT) and the value of itself, we only need to index the last $n$ cross-attention weights $a[a_1, a_2, ..., a_n]$, where $a_i$ represent the importance of the $i$-th operations. Finally, the operation associated with the largest attention weight for each layer is selected to construct the target network.

### 3.4 IN-DEPTH ANALYSIS

**Fourier Analysis.** The self-attention layer actually acts like low-pass filter with the aim of reducing high-frequency signals. Given the indicator token (IT) query vector $q_{it}$, a sequence of operational token key vectors $K = \{k_1, k_2, ..., k_n\}$, and the corresponding value vectors $V = \{v_1, v_2, ..., v_n\}$. The indicator token (IT) output of self-attention is:

$$o(q_{it}) = \frac{1}{h(q_{it})} \sum_{j=1}^{n} \exp(q_{it} \cdot k_j)v_j \tag{7}$$

where $h(q_{it}) = \sum_{j=1}^{n} \exp(q_{it} \cdot k_j)$. Further, we can rewrite Eq.(7) as follows:

$$
\begin{aligned}
o(q_{it}) &= \frac{1}{h'(q_{it})} \sum_{j=1}^{n} \exp(-\frac{1}{2} \|q_{it} - k_j\|^2) v_j \\
&= \frac{1}{h'(q_{it})} \int \exp(-\frac{1}{2} \|q_{it} - k\|^2)(\sum_{j=1}^{n} \delta(k - k_j) v_j) \mathrm{d}k \qquad (8) \\
&= \frac{1}{h'(q_{it})} G(q_{it}; 1) * S(q_{it}; K, V)
\end{aligned}
$$

where $h'(q_{it}) = \sum_{j=1}^{n} \exp(-\frac{1}{2}\|q_{it} - k_j\|^2)$ is the normalized factor, $*$ denotes the high-dimensional convolution, $G(\cdot)$ is the Gaussian kernel, and $S(\cdot)$ is high-dimensional sparse signal. Because Gaussian filter is low-pass, the indicator token output $o(q_{it})$ contains redundant information based on Shannon sampling theorem (Shannon, 1949). In other words, not all operational tokens are equally important, but the one with the largest attention weight contributes most to network performance.

**Complexity Analysis.** Compared with the complexity of $O(n^2 d)$ in the classical ViT, our method can be approximated to $O(d)$ under the condition of $n^2 \ll d$. Because $n$ generally does not exceed ten, and $d$ can be hundreds or thousands in NAS. Therefore, the search efficiency is not significantly impacted by the calculation of attention. As shown in Table 1, GPU-memory overhead is not much increased than that of the baseline method.

Table 1: Comparison of the computation complexity and GPU-Memory overhead with the baseline. The hyperparameters are all the same for fair comparison.

| Complexity | | GPU-Memory | | | |
|---|---|---|---|---|---|
| IT-NAS | ViT | IT-NAS | DARTS | IT-NAS | SPOS |
| $O(d)$ | $O(n^2 d)$ | 9.0GB | 8.5GB | 9.4GB | 7.8GB |

## 4 EXPERIMENTS

We firstly search on two popular search spaces, including DARTS search space on CIFAR-10 dataset (Krizhevsky et al., 2009), and ProxylessNAS search space on ImageNet dataset (Krizhevsky et al., 2017). Moreover, we also evaluate the robustness and effectiveness of IT-NAS on three benchmark of NAS-Bench-201, S1-S4 and NAS-Bench-1Shot1. Details are in Appendix A.1, A.2, A.5.

### 4.1 RESULTS IN DARTS SEARCH SPACE

Unlike DARTS (Liu et al., 2018b) that performs bi-level optimization by alternately updating architecture parameters and network weights, IT-NAS only need to train the super-network along with the Lite-Transformer module on half of the CIFAR-10 training dataset. The search process merely elapses 6 hours in total on Tesla V100 GPU. After the training convergence, we propagate the other half of the CIFAR-10 training dataset as validation dataset to compute the self-attention weights. Then we derive the optimal operation on each edge of the normal cell and reduction cell according to Eq.(6). To evaluate the performance, the target network consisting of 20 cells with initial channel size of 36 is trained on the whole training dataset from scratch. The details of super-network training and target network retraining settings are in Appendix A.3.

As shown in Table 2, we can see that our IT-NAS achieves state-of-the-art performance compared with other methods in the DARTS search space. We report the average results of 3 independent runs with different random seeds to test the effectiveness and stability of our method. Our approach achieves the average test error of 2.41% with the standard deviation of 0.02 on CIFAT-10 dataset, demonstrating IT-NAS is very stable. Moreover, the best result reaches the error rate of 2.39%, outperforming all the other methods. We also transfer the cells searched on CIFAR-10 to other datasets for evaluation. As a result, IT-NAS achieves the accuracy of 83.30%±0.06 on CIFAR-100 by validating the three searched cells in Figure 15. Moreover, IT-NAS achieves the Top-1 Accuracy of 75.5% on ImageNet by validating the searched Cell_2 in Figure 15.

### 4.2 RESULTS IN PROXYLESSNAS SEARCH SPACE

We directly search on ImageNet dataset to evaluate our method in the ProxylessNAS (w/o SE module) search space. Following PC-DARTS (Xu et al., 2019), we randomly sample two subsets from

Table 2: Search results on DARTS search space and comparison with other state-of-the-art methods. We report the average results for three independent runs with different initial random seeds. 'C10', 'C100', 'IMN' denotes CIFAR-10, CIFAR-100 and ImageNet, respectively.

| Methods | Test Error(%) | | | Params(M) | | | Search Cost | Search |
|---|---|---|---|---|---|---|---|---|
| | C10 | C100 | IMN | C10 | C100 | IMN | (GPU-days) | Algorithm |
| NASNet-A (Zoph et al., 2018) | 2.65 | N/A | 26.0 | 3.3 | N/A | 5.3 | 1800 | RL |
| AmoebaNet-A (Real et al., 2019) | 3.34±0.06 | N/A | 25.5 | 3.2 | N/A | 5.1 | 3150 | EA |
| AmoebaNet-B (Real et al., 2019) | 2.55±0.05 | N/A | 26.0 | 2.8 | N/A | 5.3 | 3150 | EA |
| PNAS (Liu et al., 2018a) | 3.41±0.09 | N/A | 25.8 | 3.2 | N/A | 5.1 | 225 | SMBO |
| ENAS (Pham et al., 2018) | 2.89 | N/A | N/A | 4.6 | N/A | N/A | 0.5 | RL |
| DARTS (1st order) (Liu et al., 2018b) | 3.00±0.14 | 17.76 | N/A | 3.3 | 3.3 | N/A | 0.4 | Gradient |
| DARTS (2nd order) (Liu et al., 2018b) | 2.76±0.09 | 17.54 | 26.7 | 3.3 | 3.3 | 4.7 | 1.0 | Gradient |
| SNAS (Xie et al., 2018) | 2.85±0.02 | N/A | 27.3 | 2.8 | N/A | 4.3 | 1.5 | Gradient |
| GDAS (Dong & Yang, 2019) | 2.93 | 18.38 | 26.0 | 3.4 | 3.4 | 5.3 | 0.21 | Gradient |
| BayesNAS (Zhou et al., 2019) | 2.81±0.04 | N/A | 26.5 | 3.4 | N/A | 3.9 | 0.2 | Gradient |
| Robust-DARTS (Zela et al., 2020) | 2.95±0.21 | 18.01±0.26 | N/A | N/A | N/A | N/A | 1.6 | Gradient |
| PC-DARTS (Xu et al., 2019) | 2.57±0.07 | N/A | 25.1 | 3.6 | N/A | 5.3 | 0.1 | Gradient |
| DATA (Chang et al., 2019) | 2.59 | N/A | 24.9 | 3.4 | N/A | 5.0 | 1 | Gradient |
| FairDARTS (Chu et al., 2020) | 2.54±0.05 | N/A | 24.9 | 3.32±0.46 | N/A | 5.0 | 0.4 | Gradient |
| SDARTS-ADV (Chen & Hsieh, 2020) | 2.61±0.02 | N/A | 25.6 | 3.3 | N/A | 6.1 | 1.3 | Gradient |
| DARTS+PT (Wang et al., 2021a) | 2.61±0.08 | N/A | 25.5 | 3.0 | N/A | 4.7 | 0.8 | Gradient |
| BaLeNAS (Zhang et al., 2022) | 2.50±0.07 | 16.84 | 25.0 | 3.82 | N/A | N/A | 0.6 | Gradient |
| IT-NAS (avg.) | 2.41±0.02 | 16.70±0.06 | N/A | 3.67±0.40 | 3.93±0.27 | N/A | 0.25 | Gradient |
| IT-NAS (best) | **2.39** | **16.63** | **24.5** | 3.54 | 3.59 | 5.5 | 0.25 | Gradient |

Table 3: Search results on ProxylessNAS (w/o SE module) search space and comparison with other state-of-the-art methods. ‡ denotes the search cost includes the additional subnet searching with the evolutionary algorithm.

| Methods | Test Err. (%) | | Params | FLOPs | Search Cost | Search |
|---|---|---|---|---|---|---|
| | Top-1 | Top-5 | (M) | (M) | (GPU-days) | Algorithm |
| MnasNet-92 (Tan et al., 2019) | 25.2 | 8.0 | 4.4 | 388 | 2000 | RL |
| NASNet-A (Zoph et al., 2018) | 26.0 | 8.4 | 5.3 | 564 | 1800 | RL |
| AmoebaNet-C (Real et al., 2019). | 24.3 | 7.6 | 6.4 | 570 | 3150 | EA |
| PNAS (Liu et al., 2018a) | 25.8 | 8.1 | 5.1 | 588 | 225 | SMBO |
| FBNet-C (Wu et al., 2019) | 25.1 | 7.9 | 4.4 | 375 | 9 | Gradient |
| ProxylessNAS(GPU) (Cai et al., 2018) | 24.9 | 7.5 | 7.1 | 465 | 8.3 | Gradient |
| SPOS (Guo et al., 2020) | 25.2 | N/A | 5.4 | 472 | 11‡ | Evolution |
| FairNAS-A (Chu et al., 2021b) | 24.7 | 7.8 | 4.6 | 388 | 16‡ | Evolution |
| RLNAS (Zhang et al., 2021) | 24.4 | 7.4 | 5.3 | 473 | N/A | Evolution |
| IT-NAS | **24.1** | **7.3** | 5.2 | 591 | **4** | Gradient |

1.3M training dataset of ImageNet, with 10% and 2.5% images as training and validation dataset, respectively. In particular, we train the super-network by uniformly sampling single paths following SPOS (Guo et al., 2020) on 8 Tesla V100 GPUs with a total batch size of 512 except for 240 epochs. The training process elapses 12 hours totally. After that, the optimal sub-network is derived according to the self-attention weights on each searchable layer by propagating the validation dataset once.

We restrict the mobile setting to under 600M FLOPs for fair comparison with other methods. The target network is retrained from scratch on the whole ImageNet training dataset for 240 epochs with the batch size of 1024 on 8 Tesla V100 GPUs. From Table 3, we can see that IT-NAS achieves the best performance on ImageNet. Besides, the search cost is also the lowest compared with other evolutionary algorithm-based methods, indicating that our approach is effective and efficient. The retrained architecture with SE module in ProxylessNAS space is summarized in Appendix A.6

## 4.3 RESULTS IN BENCHMARK SEARCH SPACE

**S1-S4.** We also conduct experiments on the reduced search spaces S1-S4 introduced by Robust-DARTS (Zela et al., 2020). As shown in Table 4, compared with other methods to regularize architecture parameters, IT-NAS is able to consistently select the optimal architectures with the impressive performance on all three datasets of different search spaces. This once again confirms the effectiveness of self-attention weights in indicating the importance of candidate operations.

Table 4: Comparison in the reduced search spaces S1-S4 and 3 datasets. The results follow the setting of RobustDARTS (Zela et al., 2020) where CIFAR-10 models have 20 layers and 36 initial channels except that S2 and S4 have 16 initial channels, CIFAR-100 and SVHN models have 8 layers and 16 initial channels. The best is underlined and in bold face, the second best is in bold.

| Benchmark | | DARTS | R-DARTS | | DARTS | | SDARTS-RS | DARTS+PT | IT-NAS |
|---|---|---|---|---|---|---|---|---|---|
| | | | DP | L2 | ES | ADA | | | |
| C10 | S1 | 3.84 | 3.11 | 2.78 | 3.01 | 3.10 | **2.78** | 3.50 | **2.57** |
| | S2 | 4.85 | 3.48 | 3.31 | **3.26** | 3.35 | 3.33 | N/A | **3.11** |
| | S3 | 3.34 | 2.93 | 2.51 | 2.74 | 2.59 | 2.53 | **2.49** | **2.43** |
| | S4 | 7.20 | 3.58 | **3.56** | 3.71 | 4.84 | 4.84 | N/A | **3.36** |
| C100 | S1 | 29.46 | 25.93 | 24.25 | 28.37 | 24.03 | **23.51** | 24.48 | **23.31** |
| | S2 | 26.05 | 22.30 | **22.24** | 23.25 | 23.52 | 22.28 | 23.16 | **20.58** |
| | S3 | 28.90 | 22.36 | 23.99 | 23.73 | 23.37 | **21.09** | 22.03 | **20.76** |
| | S4 | 22.85 | 22.18 | 21.94 | 21.26 | 23.20 | 21.46 | **20.80** | **20.69** |
| SVHN | S1 | 4.58 | 2.55 | 4.79 | 2.72 | 2.53 | **2.35** | 2.62 | **2.44** |
| | S2 | 3.53 | 2.52 | 2.51 | 2.60 | 2.54 | **2.39** | 2.53 | **2.36** |
| | S3 | 3.41 | 2.49 | 2.48 | 2.50 | 2.50 | **2.36** | 2.42 | **2.38** |
| | S4 | 3.05 | 2.61 | 2.50 | 2.51 | 2.46 | 2.46 | **2.42** | **2.34** |

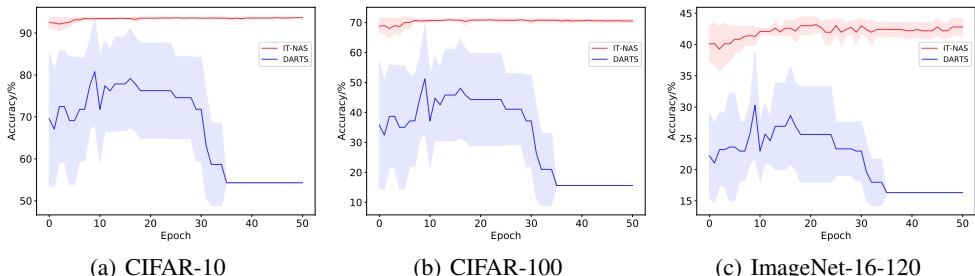

(a) CIFAR-10      (b) CIFAR-100      (c) ImageNet-16-120

Figure 2: Trajectory of test accuracy in NAS-Bench-201 on (a) cifar10, (b) cifar100, (c) Imagenet-16-120, respectively. The shaded area represents the standard deviation of six trials of experiments.

**Robustness.** To evaluate the robustness of IT-NAS, we track the performance of IT-NAS and DARTS over the search epochs on NAS-Bench-201. As plotted in Figure 2, IT-NAS achieves stable and state-of-the-art results on three datasets compared to DARTS. Whereas, the performance of DARTS is always inferior to that of our method and the standard deviation fluctuates widely as the search epochs through different trials. Because the bi-level optimization of DARTS suffers from performance collapse issues (Zela et al., 2020; Chen & Hsieh, 2020), the architecture selection process is easily disturbed by the magnitude of architecture parameters. More seriously, DARTS will downgrade the performance due to the domination of skip-connect operation at the end of the search phase (Chu et al., 2020). On the contrary, the architecture selection of IT-NAS dominated by the self-attention mechanism is more stable than architecture parameters. The results are in Table 5.

**Effectiveness.** The effectiveness of the search results can be expressed by the relationship between the architecture selection indicator and its derived architecture performance. We conduct the experiments in NAS-Bench-201 search space. We rank the selection indicator corresponding to all candidate operations on each edge, and also rank the discretization accuracy obtained by retaining the corresponding operation on this edge. The results are plotted in Figure 3, the higher the indicator rank, the more important its corresponding operation is, and so is the accuracy ranking. We also regress the relationship between the two rankings, from which we can see that the kendall-tau of IT-NAS is better than DARTS on all edges, indicating that the architecture selection criteria based on self-attention mechanism is more accurate and effective than architecture parameters.

## 4.4 VISUALIZATION ANALYSIS

Here we intuitively demonstrate whether the optimal operation selected by the self-attention is truly beneficial to the network. Let's take the super-network with six different operations on the last layer

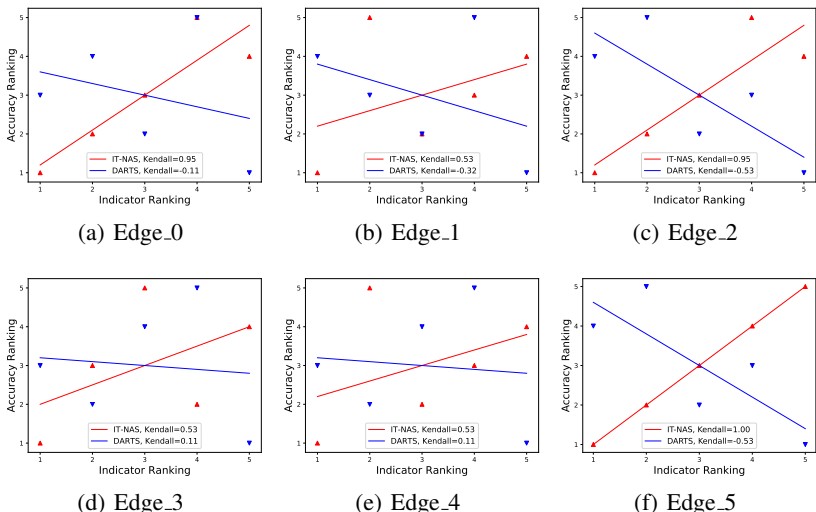

Figure 3: The relationship between indicator ranking and accuracy ranking of IT-NAS and DARTS in NAS-Bench-201 on CIFAR-10 dataset.

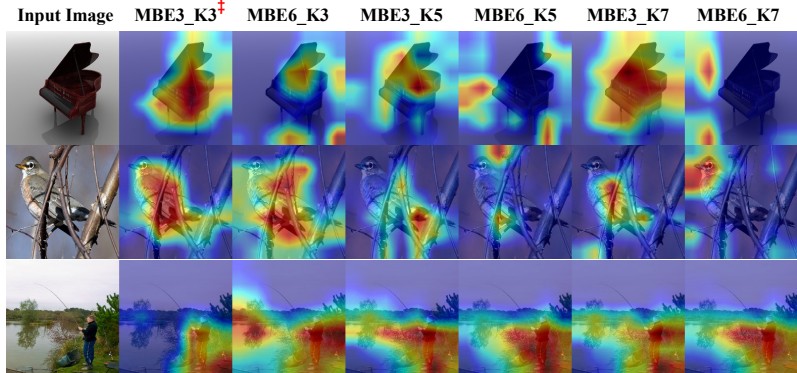

Figure 4: Visualization of the Grad-CAM (Selvaraju et al., 2017) of different candidate operations on ImageNet images. ‡ indicates the operation selected by IT-NAS in this layer.

of the ProxylessNAS search space as an example. We map the gradient information corresponding to each operation to the input image by using Grad-CAM technique (Selvaraju et al., 2017). The area with higher class activation mapping represents greater responses. As shown in Figure 4, the operation *MBE3_K3* locates the object most accurately, and the operation is also selected by IT-NAS on the last layer as shown in Figure 16. This consistency phenomenon demonstrates that the self-attention mechanism can select the most suitable operation for object perception, that in turn proves the effectiveness of our method from the perspective of gradient information.

## 5 CONCLUSION

In this paper, we propose for the first time to integrate Lite-Transformer into NAS for architecture selection. The candidate operations are regarded as tokens, and then introduce indicator token (IT) to explore the relationship between other operations and assign attention weights to which for selecting the optimal one. Comprehensive experiments demonstrate that IT-NAS is more effective compared with other architecture parameter-based or evolutionary-based architecture selection processes, and we also achieve stable results in different benchmark search spaces. This work may inspire researchers to explore more appropriate architecture selection criterion, such as borrowing from pruning or other related research, to promote the progress of the NAS field.

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

# A  APPENDIX

## A.1  DATASETS

We conduct the experiments on different image classification datasets for various search spaces.

**CIFAR-10.** The dataset contains 50K training images and 10K testing images with a fixed resolution of 32x32. The training dataset is divided into two parts, one half is used for training network weights and Lite-Transformer modules, and the other half is used as the validation dataset to forward propagation once to obtain the self-attention weights of each candidate operation for selecting the final network architecture.

**CIFAR-100.** The dataset has the same number of images as CIFAR-10 but is more categorized into 100 fine-grained classes. There are 500 training images and 100 testing images per class.

**ImageNet.** The dataset has 1.28M training images and 50K validation images with 1000 object categories. We sample 10% of the training datasets to train the super-network weights as well as the Lite-Transformer module weights, and another 2.5% training datasets to select the final architecture based on the self-attention weights.

**ImageNet-16-120.** The dataset down-samples the original ILSVRC2012 ImageNet to 16x16 resolution and only selects the first 120 categories.

**SVHN.** The dataset is a digit classification benchmark dataset that contains around 600,000 32×32 RGB images of printed digits (from 0 to 9) cropped from pictures of house number plates. SVHN has three sets: 73257 digits for training, 26032 digits for testing and an extra set with 531,131 images that are less difficult and can be used for helping with the training process.

## A.2  SEARCH SPACES

We experiment on four popular search spaces, including DARTS search space, ProxylessNAS search space, NAS-Bench-201 search space, and S1-S4.

**DARTS Search Space.** The cell-based search space aims to search for the normal cell and reduction cell. Each cell is defined as a directed acyclic graph (DAG) consisting of an ordered sequence of $N$ nodes. The edge between two nodes is mixed up by the searchable candidate operations, including {*sep-conv-3×3, sep-conv-5×5, dil-conv-3×3, dil-conv-5×5, avg-pool-3×3, max-pool-3×3, identity and none*}.

**ProxylessNAS Search Space.** The chain-style search space defines the network macro architecture and directly searches for appropriate operations at each layer. There are a total of 21 searchable layers, and each layer includes 6 candidate bottleneck blocks with different kernel size {3, 5, 7} and expansion ratio {3, 6}. Besides, the network depth can also be scaled depending on whether a skip-connect operation is selected for each layer.

**NAS-Bench-201 Search Space.** It is also the cell-based search space, but it only needs to search for the normal cell and maintain the reduction cell as a residual block with a stride of two. There are 4 nodes in the normal cell, resulting in 6 searchable edges, each of which contains 5 candidate operations, including {*nor-conv-1×1, nor-conv-3×3, avg-pool-3×3, identity and none*}.

**S1-S4.** The search space is the reduced original DARTS search space. Specifically, S1 is a pre-optimized space with two different candidate operations on each edge. S2 has the candidate operations of {*sep-conv-3×3, identity*} per edge. S3 has the candidate operations of {*sep-conv-3×3, identity, none*} per edge. S4 has the candidate operations of {*sep-conv-3×3, noise*} per edge.

**NAS-Bench-1Shot1.** The Benchmark splits NAS-Bench-101 into search space 1, search space 2 and search space 3. The three search spaces contain 6240, 29160, 363648 architectures respectively. Each architecture is consisted of three stacked blocks with max-pooling in between. Each block contains three searchable cells, and each of cell includes nine nodes.

Table 5: Search results on NAS-bench-201. We report the average performance for six independent runs of searching. "Optimal" indicates the highest accuracy for each dataset on NAS-Bench-201.

| Methods | CIFAR-10 | | CIFAR-100 | | ImageNet-16-120 | |
|---|---|---|---|---|---|---|
| | validation | test | validation | test | validation | test |
| Optimal | 91.61 | 94.37 | 73.49 | 73.51 | 46.77 | 47.31 |
| RSPS | 80.42±3.58 | 84.07±3.61 | 52.12±5.55 | 52.31±5.77 | 27.22±3.24 | 26.28±3.09 |
| DARTS | 39.77±0.00 | 54.30±0.00 | 15.03±0.00 | 15.61±0.00 | 16.43±0.00 | 16.32±0.00 |
| GDAS | 89.89±0.08 | 93.61±0.09 | 71.34±0.04 | 70.70±0.30 | 41.59±1.33 | 41.71±0.98 |
| SETN | 84.04±0.28 | 87.64±0.00 | 58.86±0.06 | 59.05±0.24 | 33.06±0.02 | 32.52±0.21 |
| ENAS | 37.51±3.19 | 53.89±0.58 | 13.37±2.35 | 13.96±2.33 | 15.06±1.95 | 14.84±2.10 |
| SNAS | 90.10±1.04 | 92.77±0.83 | 69.69±2.39 | 69.34±1.98 | 42.84±1.79 | 43.16±2.64 |
| PC-DARTS | 89.96±0.15 | 93.41±0.30 | 67.12±0.39 | 67.48±0.89 | 40.83±0.08 | 41.31±0.22 |
| DrNAS | **91.55±0.00** | **94.36±0.00** | **73.49±0.00** | **73.51±0.00** | **46.37±0.00** | **46.34±0.00** |
| IT-NAS | 90.15±0.42 | 93.65±0.04 | 69.96±1.00 | 70.31±0.66 | 43.04±0.58 | 43.88±0.98 |

## A.3 IMPLEMENTATION DETAILS

To train the super-network together with Lite-Transformer module on CIFAR-10 in DARTS search space, we use the SGD optimizer with initial learning rate 0.025, momentum 0.9 and weight decay $3 \times 10^{-4}$. The super-network is trained for 50 epochs with the batch size of 64. After deriving the final architecture, the target network consisted of 20 cells with initial channel size of 36 is trained on the whole training dataset from scratch. Specifically, we employ the SGD optimizer with initial learning rate 0.025, momentum 0.9 and weight decay $3 \times 10^{-4}$. The target network is trained for 600 epochs with the batch size of 96.

To train the super-network together with Lite-Transformer module on ImageNet in ProxylessNAS search space, we uniform sample single path from the super-network per step. Specifically, we use 8 Tesla V100 GPUs and train it for 240 epochs with a total batch size of 512. The SGD optimizer with initial learning rate 0.25, momentum 0.9 and weight decay $4 \times 10^{-5}$, and the minimum learning rate is $5 \times 10^{-4}$. The derived final network is retrained from scratch on the whole ImageNet training dataset for 240 epochs with the batch size of 1024 on 8 Tesla V100 GPUs. The SGD optimizer with initial learning rate 0.5, momentum 0.9 and weight decay $4 \times 10^{-5}$.

## A.4 RESULTS IN NAS-BENCH-201

NAS-Bench-201 is a benchmark for almost up-to-date NAS algorithms, and the diagnostic information about accuracy, loss, and parameters is accessible on three datasets including CIFAR-10, CIFAR-100, and ImageNet-16-120, respectively. We experiment with the search process on CIFAR-10 and then index the accuracy on three different datasets. We keep the hyper-parameters the same as DARTS and repeat the experiments six times with different random seeds. The results in Table 5 show that DrNAS has indeed achieved state-of-the-art results that are almost close to the global optimum. Our approach achieves comparable performance with the second best on almost datasets except obtaining the third performance on CIFAR-100.

## A.5 RESULTS IN NAS-BENCH-1SHOT1

We implement IT-NAS separately in three search spaces of NAS-Bench-1Shot1. The hyper-parameters are kept the same as DARTS. All methods, including DARTS, GDAS, PC-DARTS, ENAS, as well as IT-NAS are independently searched three times with different seeds for fair comparison. As shown in Figure 5, IT-NAS achieves almost the lowest test regret in different spaces with a smaller variance after convergence, indicating the effectiveness and robustness of IT-NAS.

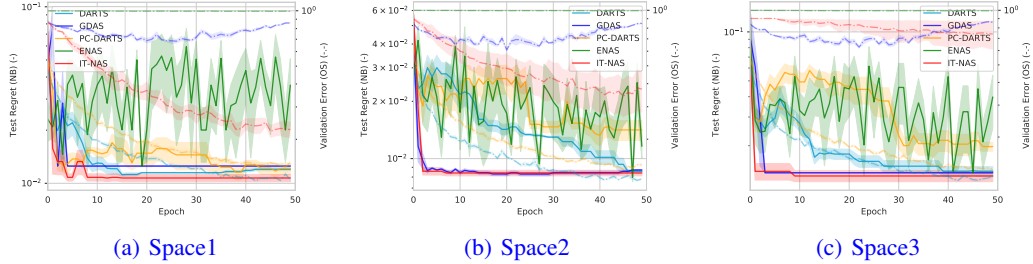

| (a) Space1 | (b) Space2 | (c) Space3 |

Figure 5: Comparison of IT-NAS with other One-Shot NAS methods on three different search spaces in NAS-Bench-1shot1. The solid lines show the anytime test regret (mean ± std), while the dashed blurred lines represent validation error (Best viewed in color).

Table 6: Comparison of the search results on ProxylessNAS (with SE module) search space on ImageNet.

| Methods | Top-1(%) | Top-5(%) |
|---------|----------|----------|
| MobileNetV3 (Howard et al., 2019) | 75.2 | N/A |
| MnasNet-A3 (Tan et al., 2019) | 76.7 | 93.3 |
| EfficientNet-B0 (Tan & Le, 2019) | 76.3 | 93.2 |
| DNA-b (Li et al., 2020) | 77.5 | 93.3 |
| BossNet-M2 (Li et al., 2021) | 77.4 | 93.6 |
| IT-NAS | **78.2** | **94.0** |

## A.6 RESULTS IN PROXYLESSNAS (WITH SE MODULE) SEARCH SPACE

We compare the searched results on ProxylessNAS search space that includes Squeeze-Excitation (SE) module. The experimental settings of retraining the search architecture are the same as DNA Li et al. (2020). Comparing the results in Table 3, the SE module can indeed promote achieving impressive results under the mobile setting. As shown in Table 6, IT-NAS achieves the best performance when compared with other methods with SE module, demonstrating the superiority of IT-NAS.

## A.7 RANKING CONSISTENCY ANALYSIS

Instead of using the discretization accuracy at convergence that proposed in DARTS-PT Wang et al. (2021a) to represent the accuracy ranking of an operation, here we leverage the best_acc definition proposed in Zero-Cost-PT (Xiang et al., 2021).

$$f_{best\_acc}(A_t, e) = \operatorname*{argmax}_{o \in \mathcal{O}_e} \max_{A_{|\varepsilon|} \in \mathcal{A}_{t,e,o}} V^*(A_{|\varepsilon|}) \tag{9}$$

where $V^*$ denotes validation accuracy of a network after full training, $A_{|\varepsilon|}$ denotes all possible fully-discretized subnetworks, $t$ is the discretization iteration of an edge $e$ with operation $o$.

We plot the relationship between best_accuracy ranking and indicator ranking of IT-NAS and DARTS in NAS-Bench-201 on CIFAR-10 dataset. The indicator can be self-attention weights of IT-NAS or architecture parameters of DARTS. As shown in Figure 6, we can see that IT-NAS achieves almost positive ranking consistency except on the Edge_2( Even so, IT-NAS also obtains a better ranking than DARTS on Edge_2.) Besides, the Kendall tau ranking is higher than DARTS on all edges, demonstrating the effectiveness of self-attention weight in evaluating the importance of candidate operations.

To compare the ranking results with Zero-Cost-PT that leverage Zero-Cost proxy to predict the performance of candidation operation, we choose to combine Zero-Cost-PT with NASWOT and SynFlow proxy to predict the score of each operation per edge. The results are plotted in Figure

7 and Figure 8. The results show that SynFlow obtains generally better accuracy ranking than NASWOT when combining with Zero-Cost-PT, but IT-NAS still achieves better ranking on most edges than Zero-PT-NASWOT and Zero-PT-SynFlow. Though zero-cost proxies are fast in assessing the importance of operations, its accuracy is inferior to the self-attention mechanism.

## A.8 MORE EMPIRICAL STUDIES ON ACCURACY RANKING

Based on A.7, we analyze the self-attention weights or architecture parameters in indicating the best_accuracy ranking. We plot the accuracy and indicator of all six edges on NAS-Bench-201 in Figure 9-Figure 14. Taking the first edge as an example, the maximum self-attention weight and the highest accuracy of IT-NAS are both on nor_conv_3x3 operation, demonstrating the effectiveness of self-attention weight in evaluating the importance of candidate operations. Whereas, the largest architecture parameter of DARTS is skip_connect, which does not contribute to the highest accuracy, so that the final derived architecture according to alpha would not obtain the satisfactory accuracy. On the other hand, the skip_connect weights are always the highest across all edges, showing that the bi-level optimization of DARTS is prone to performance collapse. The search results of IT-NAS are more stable due to the mono-level optimization.

## A.9 VISUALIZATION

Here we visualize the searched normal cells and reduction cells from three different experiments with different initial random seeds in DARTS search space. As shown in Figure 15, the test error and parameters of network based on (a)(b) are 2.39% and 3.54M respectively; the test error and parameters of network based on (c)(d) are 2.39% and 4.21M respectively; the test error and parameters of network based on (e)(f) are 2.44% and 3.27M respectively.

Moreover, as shown in Figure 16, we also visualize the searched chain-style architecture in the ProxylessNAS search space on ImageNet directly.

Figure 17, Figure 18, Figure 19 separately shows the searched cells on CIFAR-10, CIFAR-100, and SVHN in the reduced search spaces S1-S4, respectively. The corresponding accuracy of each cell can be seen in the Table 4.

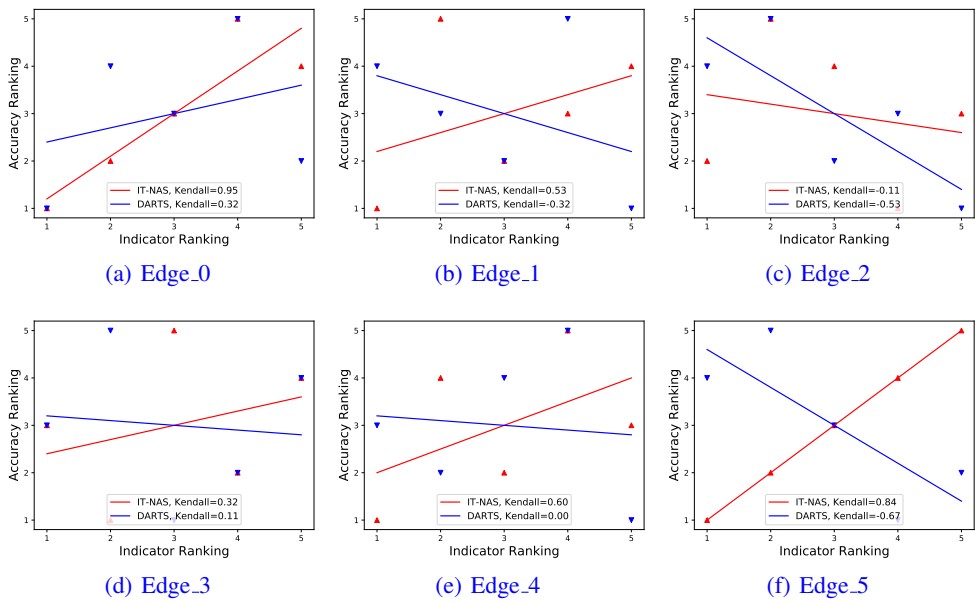

Figure 6: The relationship between indicator ranking and accuracy ranking of IT-NAS and DARTS in NAS-Bench-201 on CIFAR-10 dataset.

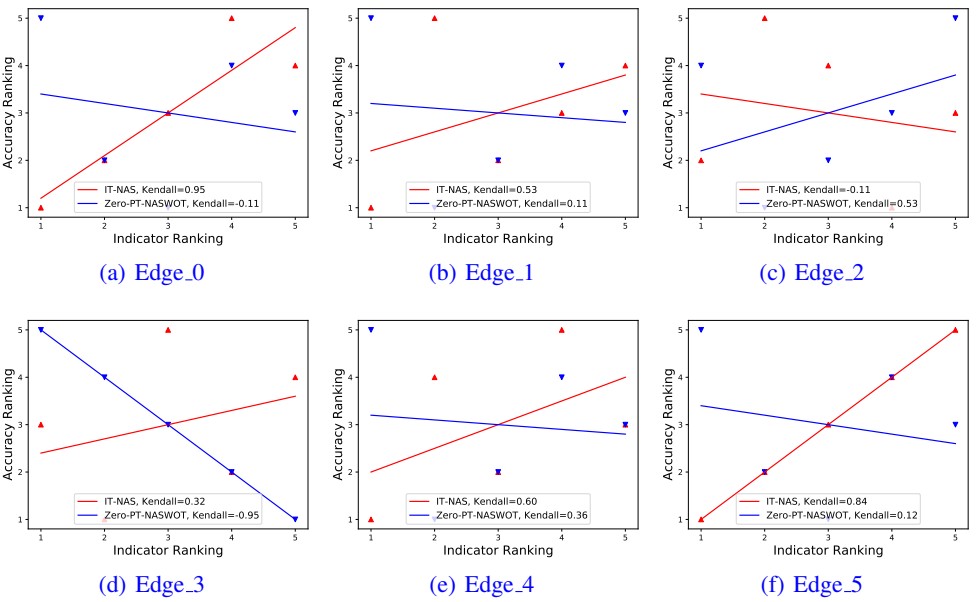

Figure 7: The relationship between indicator ranking and accuracy ranking of IT-NAS and Zero-Cost-PT with NASWOT in NAS-Bench-201 on CIFAR-10 dataset.

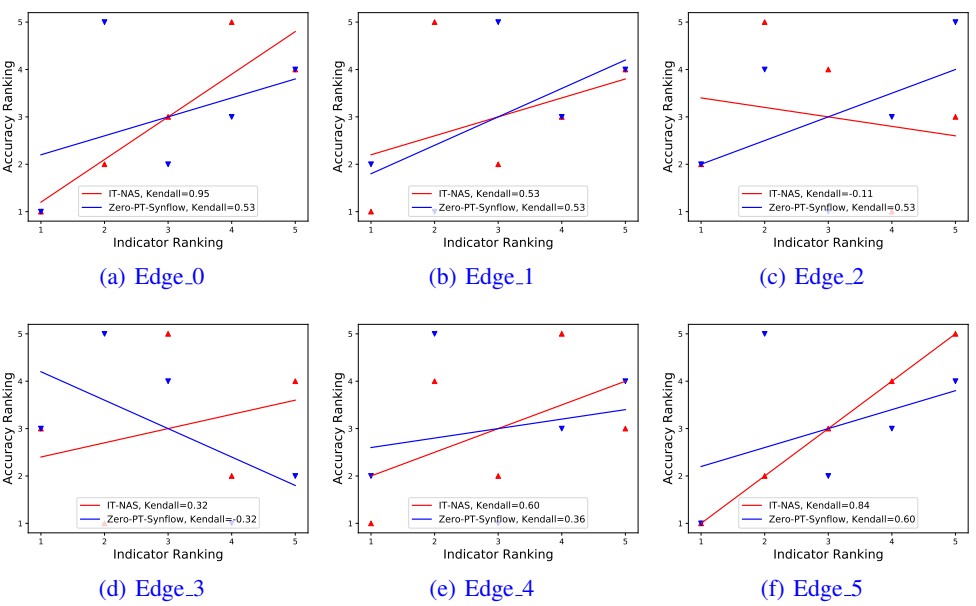

Figure 8: The relationship between indicator ranking and accuracy ranking of IT-NAS and Zero-Cost-PT with SynFlow in NAS-Bench-201 on CIFAR-10 dataset.

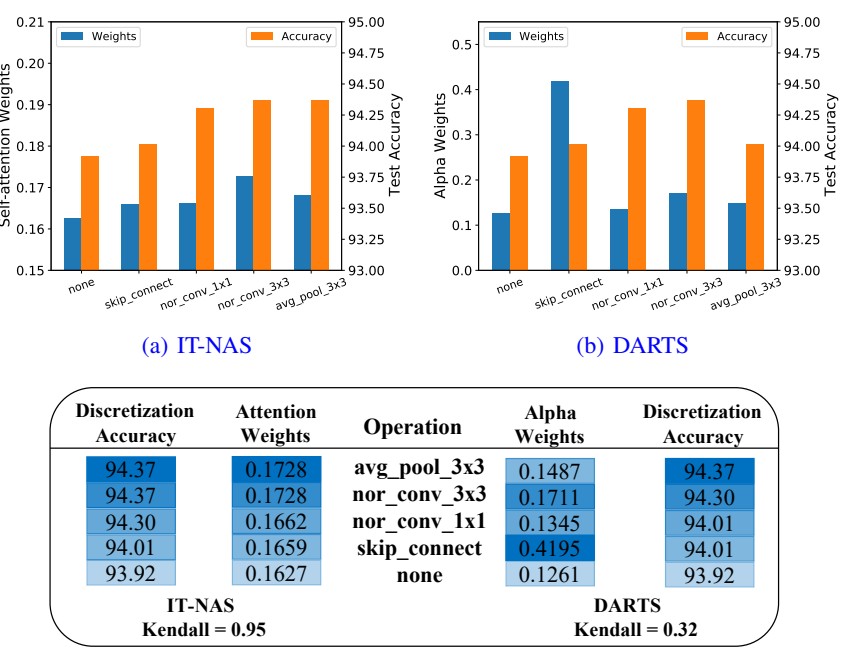

(a) IT-NAS

(b) DARTS

| Discretization Accuracy | Attention Weights | Operation | Alpha Weights | Discretization Accuracy |
|---|---|---|---|---|
| 94.37 | 0.1728 | avg_pool_3x3 | 0.1487 | 94.37 |
| 94.37 | 0.1728 | nor_conv_3x3 | 0.1711 | 94.30 |
| 94.30 | 0.1662 | nor_conv_1x1 | 0.1345 | 94.01 |
| 94.01 | 0.1659 | skip_connect | 0.4195 | 94.01 |
| 93.92 | 0.1627 | none | 0.1261 | 93.92 |
| **IT-NAS** Kendall = 0.95 | | | **DARTS** Kendall = 0.32 | |

(c) The ranking of attention weights or alpha weights against discretization accuracy.

Figure 9: Comparison of self-attention weights and architecture parameters on measuring the importance of candidate operations both on the 1st edge of the cell in NAS-Bench-201.

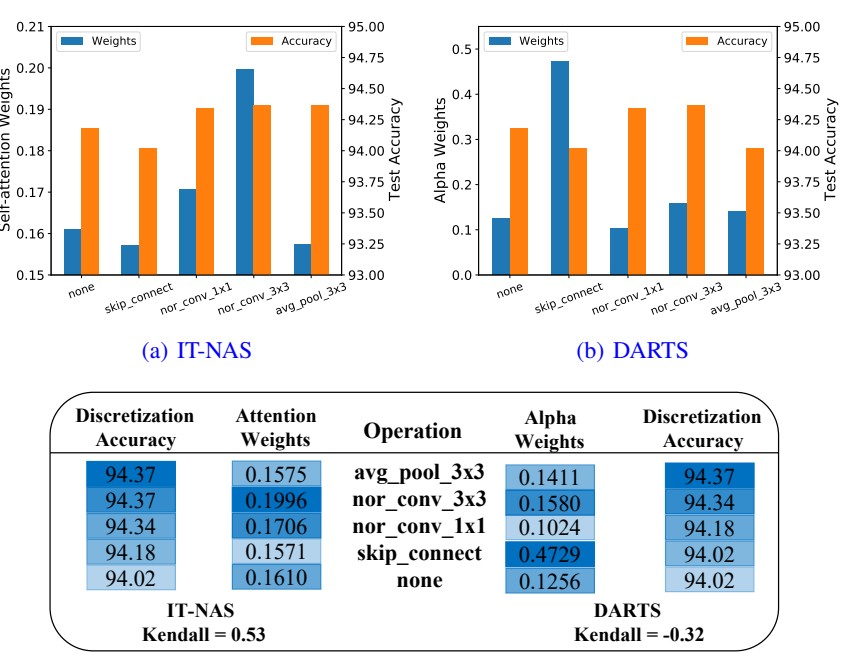

(a) IT-NAS

(b) DARTS

| Discretization Accuracy | Attention Weights | Operation | Alpha Weights | Discretization Accuracy |
|---|---|---|---|---|
| 94.37 | 0.1575 | avg_pool_3x3 | 0.1411 | 94.37 |
| 94.37 | 0.1996 | nor_conv_3x3 | 0.1580 | 94.34 |
| 94.34 | 0.1706 | nor_conv_1x1 | 0.1024 | 94.18 |
| 94.18 | 0.1571 | skip_connect | 0.4729 | 94.02 |
| 94.02 | 0.1610 | none | 0.1256 | 94.02 |
| **IT-NAS** Kendall = 0.53 | | | **DARTS** Kendall = -0.32 | |

(c) The ranking of attention weights or alpha weights against discretization accuracy.

Figure 10: Comparison of self-attention weights and architecture parameters on measuring the importance of candidate operations both on the 2nd edge of the cell in NAS-Bench-201.

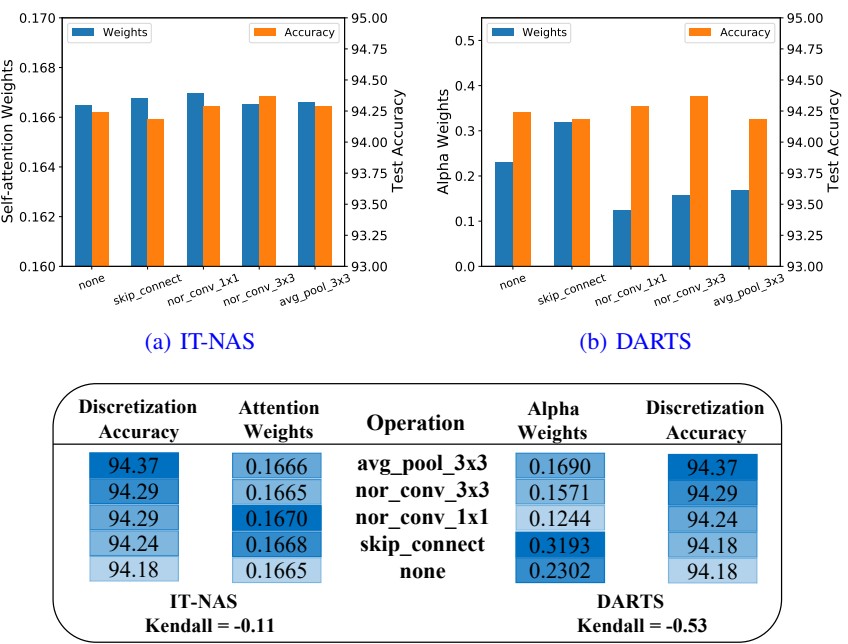

(a) IT-NAS

(b) DARTS

| Discretization Accuracy | Attention Weights | Operation | Alpha Weights | Discretization Accuracy |
|---|---|---|---|---|
| 94.37 | 0.1666 | avg_pool_3x3 | 0.1690 | 94.37 |
| 94.29 | 0.1665 | nor_conv_3x3 | 0.1571 | 94.29 |
| 94.29 | 0.1670 | nor_conv_1x1 | 0.1244 | 94.24 |
| 94.24 | 0.1668 | skip_connect | 0.3193 | 94.18 |
| 94.18 | 0.1665 | none | 0.2302 | 94.18 |
| IT-NAS Kendall = -0.11 | | | DARTS Kendall = -0.53 | |

(c) The ranking of attention weights or alpha weights against discretization accuracy.

Figure 11: Comparison of self-attention weights and architecture parameters on measuring the importance of candidate operations both on the 3rd edge of the cell in NAS-Bench-201.

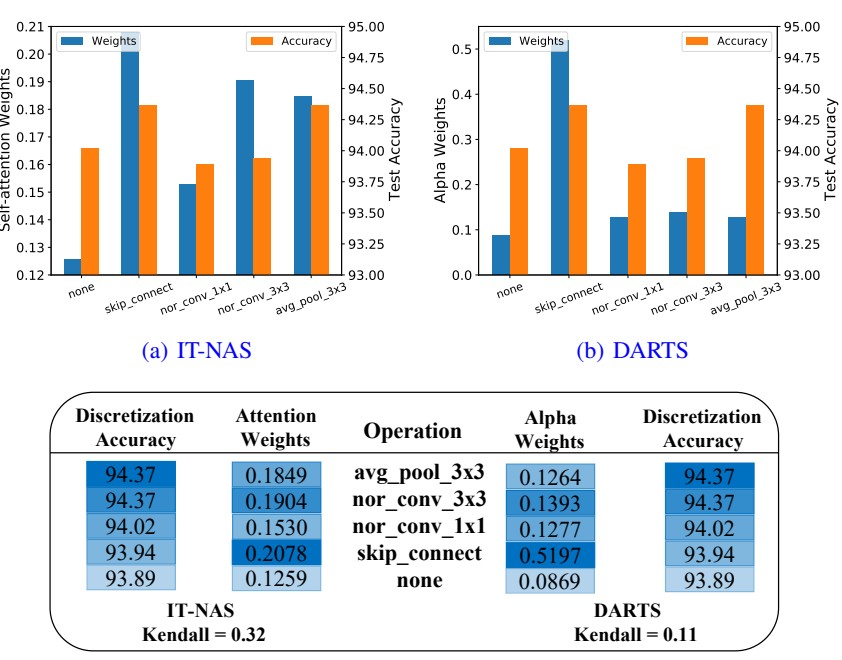

(a) IT-NAS

(b) DARTS

| Discretization Accuracy | Attention Weights | Operation | Alpha Weights | Discretization Accuracy |
|---|---|---|---|---|
| 94.37 | 0.1849 | avg_pool_3x3 | 0.1264 | 94.37 |
| 94.37 | 0.1904 | nor_conv_3x3 | 0.1393 | 94.37 |
| 94.02 | 0.1530 | nor_conv_1x1 | 0.1277 | 94.02 |
| 93.94 | 0.2078 | skip_connect | 0.5197 | 93.94 |
| 93.89 | 0.1259 | none | 0.0869 | 93.89 |
| IT-NAS Kendall = 0.32 | | | DARTS Kendall = 0.11 | |

(c) The ranking of attention weights or alpha weights against discretization accuracy.

Figure 12: Comparison of self-attention weights and architecture parameters on measuring the importance of candidate operations both on the 4th edge of the cell in NAS-Bench-201.

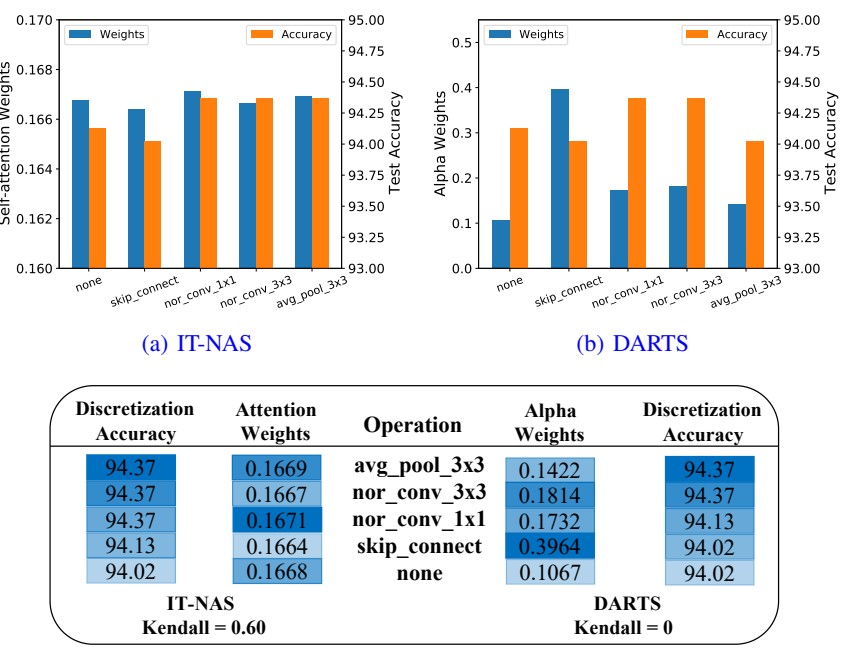

(c) The ranking of attention weights or alpha weights against discretization accuracy.

Figure 13: Comparison of self-attention weights and architecture parameters on measuring the importance of candidate operations both on the 5th edge of the cell in NAS-Bench-201.

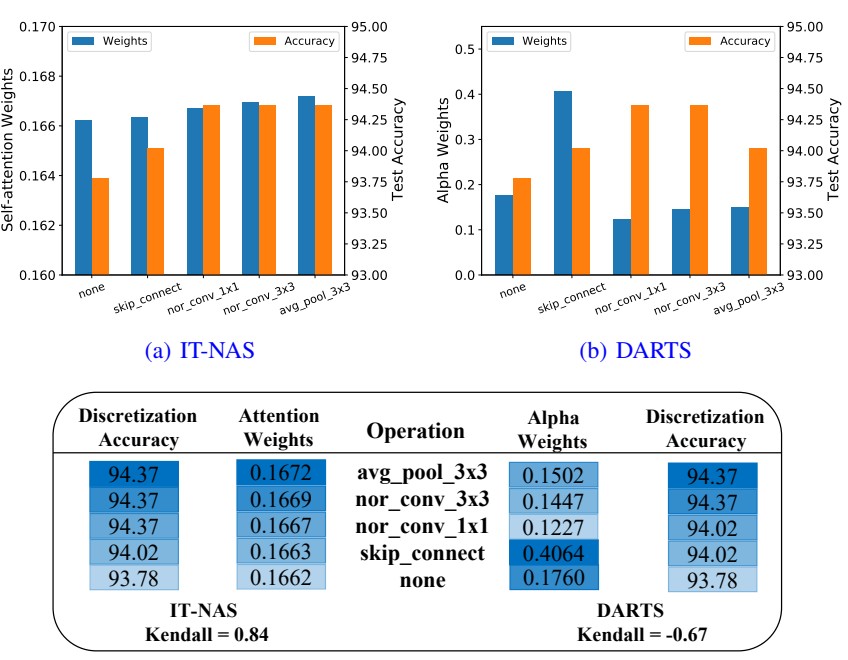

(c) The ranking of attention weights or alpha weights against discretization accuracy.

Figure 14: Comparison of self-attention weights and architecture parameters on measuring the importance of candidate operations both on the 6th edge of the cell in NAS-Bench-201.

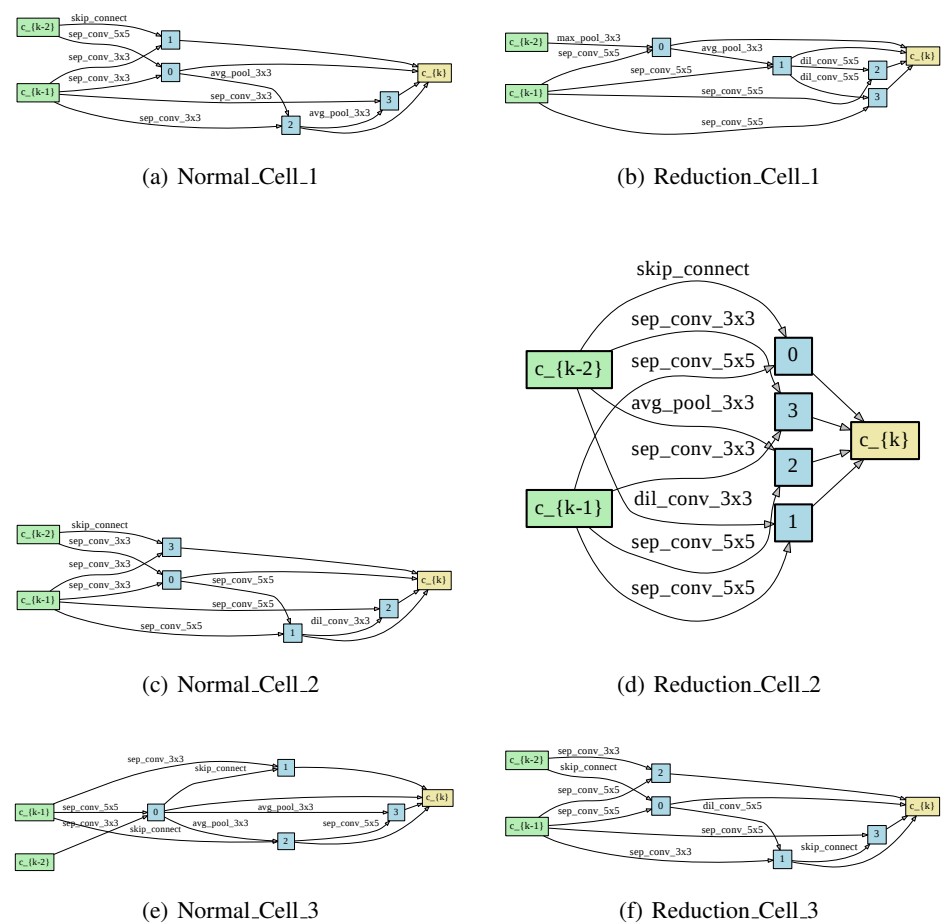

Figure 15: The searched normal cells and reduction cells on CIFAR-10 in DARTS search space.

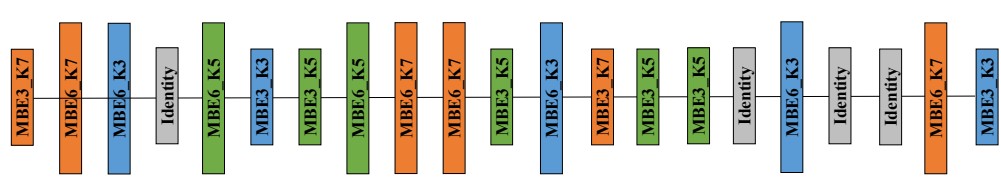

Figure 16: The searched chain-style architecture on ImageNet in ProxylessNAS search space.

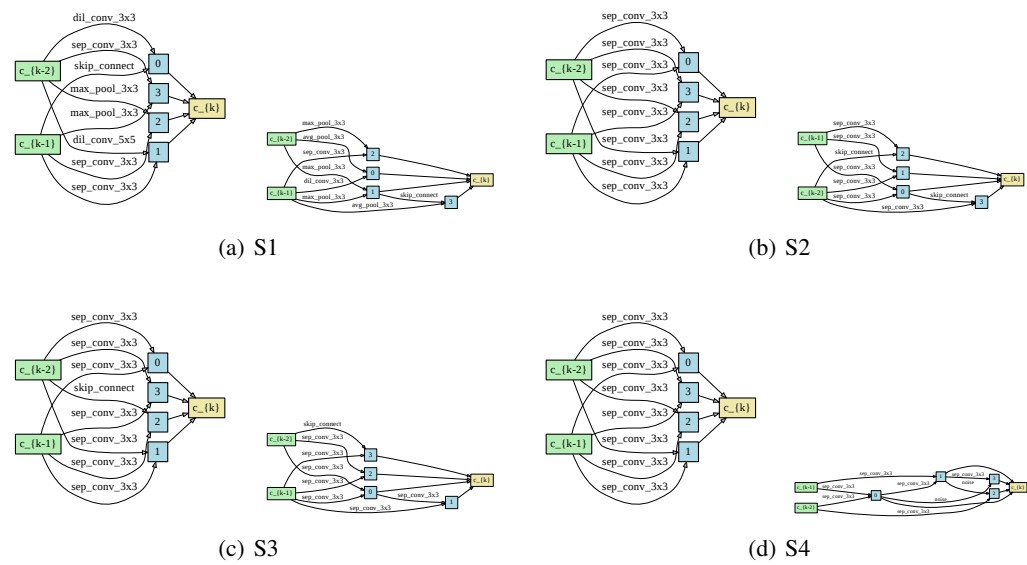

Figure 17: IT-NAS best cells (paired in normal and reduction) on CIFAR-10 in reduced search spaces of RobustDARTS.

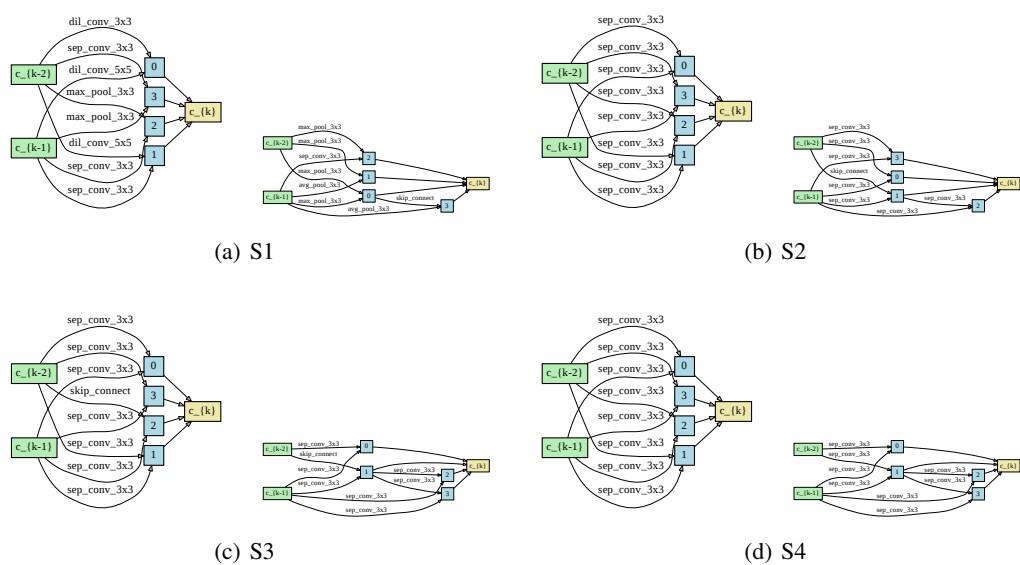

Figure 18: IT-NAS best cells (paired in normal and reduction) on CIFAR-100 in reduced search spaces of RobustDARTS.

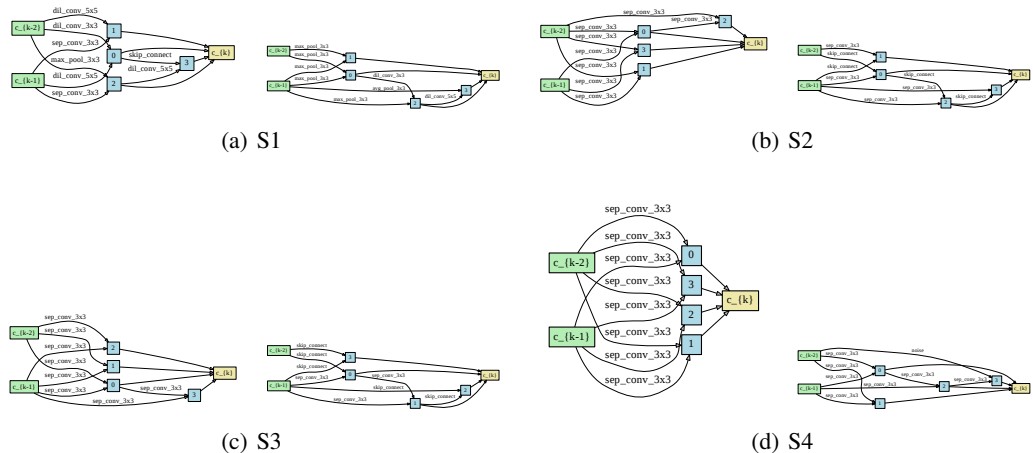

(a) S1

(b) S2

(c) S3

(d) S4

Figure 19: IT-NAS best cells (paired in normal and reduction) on SVHN in reduced search spaces of RobustDARTS.

