# OpenReview forum: "IT-NAS: Integrating Lite-Transformer into NAS for Architecture Seletion"
_ICLR.cc/2023/Conference — Submitted to ICLR 2023_

### Official Review · Reviewer_3FDx · 2022-10-27

**Confidence:** 3
**Clarity, Quality, Novelty And Reproducibility:** Overall, this paper is of good qualit…
**Correctness:** 3
**Technical Novelty And Significance:** 3
**Empirical Novelty And Significance:** 3
**Recommendation:** 6

**Strength And Weaknesses:**

strengths:
1. This paper is generally well-written and easy to follow.
2. To the best of my knowledge, this paper is the first to introduce a lite transformer for selecting the final architecture in NAS, which is simple yet works well.
2. Empirical results in this paper are attractive and have achieved SOTA.

weaknesses/questions:
1. Zit in eq.4 and W in eq.5 are not well-explained. Specifically, how is Zit used during the model training of supernet and what's W in eq.5?
2. There lacks a theoretical justification for why a lite transformer is a better alternative to the architecture parameters in DARTS.
3. How many additional parameters have been introduced to NAS compared with the architecture parameters? Will it affect the bi-level optimization (e.g., make the optimization harder) due to an increasing number of parameters?
4. In addition to figure 3, more empirical studies on why a lite transformer can achieve a better characterization of accuracy ranking perhaps can make this paper more attractive and inspiring.

**Summary Of The Paper:**

Existing training-based NAS algorithms typically select the final architecture that achieves the largest architecture parameters. This however can be unstable and unreliable in practice because of the limited representation ability of these architecture parameters. To this end, this paper introduces the self-attention mechanism (lite-transformer) to better characterize the importance of different operations in NAS, which has been supported by the empirical results and ablation studies in this paper.

**Summary Of The Review:**

In general, this is an interesting paper and I hope the authors can address my concerns during the rebuttal period.

---

### Official Review · Reviewer_CZtJ · 2022-10-28

**Confidence:** 4
**Correctness:** 3
**Technical Novelty And Significance:** 2
**Empirical Novelty And Significance:** 3
**Recommendation:** 3

**Clarity, Quality, Novelty And Reproducibility:**


The authors emphasized that they are the first to incorporate transformer into NAS selection. But what is the real motivation other than this has never been done in the literature? Why using a self-attention mechanism can extract the relationship among them? This remains unanswered.


**Strength And Weaknesses:**

# Strength:

The paper proposes to explore self-attention to select the superior operations of NAS.

# Weaknesses:

As claimed in the introduction, the motivation to incorporate transformer into differentiable architecture search pipeline is to improve the architecture selection process in differentiable NAS algorithms. So ideally there should be two aspect that this IT-NAS is truly superior:

Showing the search results is superior on commonly benchmarked dataset, such as in DARTS+PT original paper. Recently, there is a zero-cost PT showing a best architecture of 2.43 top1 error on CIFAR-10, while only using 0.018 hours search cost. Compared to this transformer based one, I do not see much reason why involving a transformer can improve the result. And the figure 3 is showing their architecture ranking is superior than the original DARTS in 2022 is too weak to be accepted to the top-tier conference.
Results on ProxylessNAS search space are not state-of-the-art, e.g., in BossNAS, the architecture search results on proxylessNAS space achieve 77.4% in 2021 under the 600M FLOPs constraint. I hope the authors can consider doing their experiment in recent literature rather than comparing to those out-dated methods.

In one word, this paper claim to discover a better architecture ranking in differentiable architecture search domain, I suggest the authors take a closer look at these papers and provide a detailed comparison to show IT-NAS truly surpasses these baselines.

[a] Zero-Cost Proxies Meet Differentiable Architecture Search, Xiang et al.

[b] BossNAS: Exploring Hybrid CNN-transformers with Block-wisely Self-supervised NAS, Li et al.

**Summary Of The Paper:**


This paper integrates Transformer into NAS for architecture selection by regarding each candidate operation as a patch and also introducing an additional importance indicator token (IT) to calculate cross-attention among the other operational tokens.


**Summary Of The Review:**


This method seems to apply an existing well-known approach (transformer architecture) into architecture search to address the architecture selection problem. However, compared to earlier works, like DARTS-PT, Zero-cost-PT and BossNAS, they did not provide sufficient evidence that their method can indeed generate better architecture ranking results. The only proof is they show is via the final searched architecture results, which is interesting but definitely inadequate to prove the claim is valid.

---

Update after reading the response

I would like to thank the authors to provide such a detailed and comprehensive responses to address my concerns. Before the revision, this paper suffers from two major weaknesses, marginally technical novelty that directly apply the transformer self-attention mechanism without much support of their claim, using self-attention can boost the architecture selection problem; insufficient comparison with the recent literature. After the rebuttal, several of my previous concerns are addressed, especially regarding the empirical one. IT-NAS surpasses many state-of-the-art methods in the differentiable architecture search domain.

However, as there is no borderline score, I would still maintain my score to reject this work for the following reasons.

1. Major revision needed to truly address the weakness but is not reflected in the current revision.
Many interesting results to support their claims (such as Table 5, A.4, A.6, A.7) are still in the appendix, even in their revised version. With respect, if I read this paper again without the supplementary material, I would still have the same concern. This is because reading supplemenary materials should not be mandatory to the reviewers or general readers. In the revised version, for example, Figure 2, 3, only shows the comparison between IT-NAS with the original DARTS, giving the readers a feeling that they only compared to the weakest baseline that was introduced almost four years ago. I am aware these can be fixed, but it is essentially equivalent to re-write the entire experiment section. Given such substantial changes, I would recommend this paper to go over another review cycle.

2. Incremental technical novelty to use a well-known technique, self-attention to another domain.
This work seems novel in the first screening, however, it does not change the fact that this is simple attempt to use a well known successful technique, self-attention, into another domain, differentiable architecture search. This is not wrong, but leads to incremental technical novelty.

Combining these two, I cannot recommend acceptance at this time.

---

### Official Review · Reviewer_2Hwu · 2022-10-30

**Confidence:** 4
**Correctness:** 3
**Technical Novelty And Significance:** 3
**Empirical Novelty And Significance:** Not applicable
**Recommendation:** 6

**Clarity, Quality, Novelty And Reproducibility:**

The idea presented in this paper is pretty novel and worth exploring. However, the clarity of the paper can be improved. There is also a brief section in the appendix on implementation details, but no code is provided for reproducibility.

Some specific questions / comments:

- It is not clear whether weights are shared among Lite-Transformers for different edges / layers.
- In eq.5 why do the super-net N and the Lite-Transformer T depend on the same set of weights W?
- To go from eq.7 to the first line of eq.8 an exp(-k_j^2) factor seems to be ignored?
- For the DARTS search space on ImageNet why are the results on cell_1, cell_3 not reported?
- Based on the last paragraph of section 4.3 it seems accuracy ranking of an operation (which is used in In Figure.3 to calculate correlations with predicted rankings) is defined by retaining that operation on the corresponding edge. If this is the case, why is accuracy ranking an indicator of the true ranking. It might be useful to use instead the best_acc defined in eq.1 of the paper “Zero-Cost Proxies Meet Differentiable Architecture Search” which is also available for NAS-Bench-201.
- In table.5 the highest accuracies on cifar100 should be made bold.

Removing minor typos would improve readability. Some examples are:

- Caption of Figure.1: assign attention weights to which → assign attention weights to them
- Last line of section 3.3: largest attention weight each layer → largest attention weight for each layer
- Line before “complexity analysis”: contributes mostly to network performance → contributes most to network performance
- First paragraph of section 4.1: target network consisted of 20 cells → target network consisting of 20 cells
- Second line after Table.3: by uniform sampling single paths → by uniformly sampling single paths
- Fifth line after Table.3: propagating the validation dataset once time → propagating the validation dataset once.
- Caption of Table.4: The results are follow the setting → The results follow the setting


**Strength And Weaknesses:**

Using the self-attention mechanism to rank operations is interesting, especially that the Lite-Transformer is trained jointly with the one-shot model, making the search a mono-level optimization.

However, the experiments on the NAS-Bench-201 space do not include comparisons with some well-known methods such as DARTS-PT (already included in Table.2 for comparisons on the DARTS search space) and DrNAS. Based on the results in Table.5, the method does not seem to achieve state-of-the-art performance on NAS-Bench-201 as claimed in the summary of contributions (third bullet point).

In order have a better assessment of the method and perhaps strengthen the results it would be useful to extend the experiments to other search spaces, compatible with one-shot optimization. NAS-Bench-101 which is also a tabular benchmark, would be a natural choice.

**Summary Of The Paper:**

Motivated by the fact that architecture parameters in one-shot differentiable NAS are not indicative of the true operator importance, the authors integrate the Lite-Transformer into the one-shot model, and train it jointly with the model, to predict the ranking of operators at each edge / layer.

The operations are passed into the Lite-Transformer as a sequence, together with an extra indicator token whose attention to other operational tokens determines the operator importance.

Comparison with other search strategies is performed on DARTS, ProxylessNAS, S1-S4 and NAS-Bench-201 search spaces, on most of which superior performance is achieved.

**Summary Of The Review:**

The idea is very interesting and worth exploring, and the results seem promising. However, some claims are disputable, and the presentation can be slightly improved. I would therefore rate the paper as marginally above the acceptance threshold.

---

### Official Review · Reviewer_o9WQ · 2022-11-02

**Confidence:** 5
**Correctness:** 3
**Technical Novelty And Significance:** 3
**Empirical Novelty And Significance:** 3
**Recommendation:** 6

**Clarity, Quality, Novelty And Reproducibility:**

To be honest, I reviewed this paper in NeurIPS 2022. In this re-submission, I'm happy to see significant improvements over the previous version. I acknowledge that the general presentation and performance become better, but I'm still confused about the over-claiming issue as mentioned above. I'm looking forward to communicating with other reviewers and AC.

**Strength And Weaknesses:**

[ Strength ]
+ The proposed method seems new and interesting. Search space selection/optimization is a classical topic in NAS, and the authors take advantage of popular transformers (more precisely, the attention mechanism) to measure the importance of each candidate operation. This is intuitive and straightforward.
+ In-depth analysis in Section 3.4 is important and informative. It helps to better understand the proposed method, especially the motivation of introducing the indicator token. The complexity analysis is also very important because it would be confusing that the complexity is quadratic to the length of input tokens (i.e., the number of candidate operations) without the analysis.
+ Experimental results can well support the effectiveness of the proposed method. It achieves good performance in accuracy, efficiency, and search quality.

[ For further improvements ]
In essence, the method applies the attention mechanism into NAS. It looks like that this intuitive operation is just engineering-level rather than being important to the scientific research of NAS or transformers. Also, using attention for selection is not the unique property of transformers, i.e., some other network structures may also use attention for selection. It seems that the role of transformers here is a little bit over-claimed (It is also confusing that the authors use the term "lite-transformer" as it is just the very common transformer module).

**Summary Of The Paper:**

This paper proposes to apply NAS to architecture selection in Transformers. Each candidate operation is measured by the attention mechanism that can be learned and optimized in the network training. The proposed method is insightful and interesting, and achieves competitive performance in efficiency and accuracy, as well as the search quality.


**Summary Of The Review:**

I would lean to accept the paper in terms of its improvements over the last version. Some critical points still need to be further clarified.

---

### Decision · Program_Chairs · 2023-01-20

**Decision:**

Reject

**Justification For Why Not Higher Score:**

The union of the issues mentioned above. But the paper could also be accepted if that would fit the big picture better; one reason for doing so would be good performance on a rather large set of search spaces.

**Justification For Why Not Lower Score:**

N/A

**Metareview: Summary, Strengths And Weaknesses:**

This paper integrates attention into the architecture selection step of NAS.
The problem tackled is seen as important by all reviewers (and I agree); the approach is simple and appears to work well. The authors evaluated on quite a lot of search spaces. However, the paper also has some issues.
During the rebuttal, the authors added quite a bit of material to address the issues raised, but as reviewer CZtJ put it, a major revisision would be needed to truly reflect these in the paper, rather than adding them to the supplementary information, basically as an afterthought.
Other issues include low technical novelty (attention is a well-known concept that works well in many domains) and a lack of clarification regarding the nomenclature of Lite-Transformer (it appears to be just the standard Transformer module / self-attention, which reviewer o9WQ called "overclaiming"). Reviewer 2Hwu also remarked that no code is provided for reproducibility, a comment the authors left unanswered.
Based on the union of these issues I recommend rejection, but since the work clearly has value I encourage the authors to fix these and resubmit.